# CAP-YOLO: Channel Attention Based Pruning YOLO for Coal Mine Real-Time Intelligent Monitoring

**DOI:** 10.3390/s22124331

**Published:** 2022-06-08

**Authors:** Zhi Xu, Jingzhao Li, Yifan Meng, Xiaoming Zhang

**Affiliations:** College of Electrical and Information Engineering, Anhui University of Science and Technology, Huainan 232000, China; xu1995zhi@126.com (Z.X.); ivanmeng@outlook.com (Y.M.); 2017100038@aust.edu.cn (X.Z.)

**Keywords:** channel attention mechanism, model pruning, object detection, image enhancement, YOLO

## Abstract

Real-time coal mine intelligent monitoring for pedestrian identifying and positioning is an important means to ensure safety in production. Traditional object detection models based on neural networks require significant computational and storage resources, which results in difficulty of deploying models on edge devices for real-time intelligent monitoring. To address the above problems, CAP-YOLO (Channel Attention based Pruning YOLO) and AEPSM (adaptive image enhancement parameter selection module) are proposed in this paper to achieve real-time intelligent analysis for coal mine surveillance videos. Firstly, DCAM (Deep Channel Attention Module) is proposed to evaluate the importance level of channels in YOLOv3. Secondly, the filters corresponding to the low importance channels are pruned to generate CAP-YOLO, which recovers the accuracy through fine-tuning. Finally, considering the lighting environments are varied in different coal mine fields, AEPSM is proposed to select parameters for CLAHE (Contrast Limited Adaptive Histogram Equalization) under different fields. Experiment results show that the weight size of CAP-YOLO is 8.3× smaller than YOLOv3, but only 7% lower than mAP, and the inference speed of CAP-YOLO is three times faster than that of YOLOv3. On NVIDIA Jetson TX2, CAP-YOLO realizes 31 FPS inference speed.

## 1. Introduction

Intelligent coal mine video surveillance is an important measure to ensure production safety. The pedestrians in the monitoring video are detected by AI (Artificial Intelligence) models and control the equipment or issue alarms according to the pedestrian position, which can effectively prevent the operating equipment from causing injury to workers.

The CNN (Convolutional Neural Network) has achieved remarkable success in the field of intelligent image processing, and the accuracy of image classification models based on CNNs have even surpassed that of human beings [1]. With the excellent performance of CNNs in feature extraction, various CNN-based object detection models have been proposed and used in different fields [2,3]. However, traditional object detection models are usually deployed on cloud servers due to the large demands of computing and storage resources. When intelligent analysis of monitoring video is required, surveillance video must be transmitted to cloud servers through the network. Then, the monitoring videos are analyzed by AI models in cloud servers and return the results of video analysis through the network. The whole process of cloud computing produces serious transmission latency because of the limitation of network bandwidth. Meanwhile, transmitting a large amount of surveillance video also causes serious network congestion [4]. Edge computing is proposed to decentralize intelligent computing close to the data source for avoiding transmission latency and network congestion. Therefore, deploying object detection models on embedded platforms can not only avoid the problems caused by cloud computing, but also control equipment or alarm devices in real-time according to the video analysis results. However, it is difficult to deploy AI models on edge due to the constraint of computing and storage resources of embedded platforms.

To deploy CNN models on embedded platforms, neural network compression methods have received a lot of attention from researchers. Neural network compression aims to reduce the number of parameters or calculations through model pruning, weight quantization, knowledge distillation, or other methods, to greatly improve real-time performance [5]. Model pruning improves the inference speed by removing redundant neurons [6,7,8]. The pruning approaches for CNN can be roughly divided into non-structured pruning and structured pruning. The inference speed is difficult to accelerate because of the irregular memory access of the non-structure pruned model, unless using specialized hardware or libraries [9]. Structured pruning prevents the structure of CNN by directly removing whole filters [8]. However, it is necessary to evaluate the importance of the pruned filters/channels or weights for the two pruning methods. We only focus on structured pruning in this paper.

Currently, there are various approaches to evaluate the importance of filters or channels for structured pruning [7,8,10]. Attention mechanism [11,12] is used to enhance the important information and suppress unnecessary information [13]. It was widely used in NLP (Natural Language Processing) at first, and then it has been introduced into the computer vision field [14]. Attention mechanism improves the performance of computer vision by important feature enhancement [15]. The output scale value of attention mechanism represents the enhancement value and the importance level of the features. Therefore, some researchers have designed channel attention modules for model pruning. Channel attention mechanisms evaluate the importance level of channels, and the filters corresponding to low-importance channels will be pruned [16]. However, the application of attention mechanism in pruning object detection models is rare. Moreover, the high complexity of the object detection model requires an advanced channel attention module for evaluating channel importance levels.

YOLO is a classical one-stage object detection model [2]. It has the advantages of high real-time performance and fewer parameters compared with two-stage models [17]. In order to deploy YOLO on embedded platforms, researchers have undertaken a lot of work to reduce the number of parameters and calculations [18,19,20,21]. However, how to identify redundant channels or filters is still a challenge. CLAHE is usually combined with object detection models [22] for improving detection performance. However, the lighting environments in coal mines are complex and variable, and the lighting conditions in different monitoring areas are also different. Therefore, it is necessary to set the parameters of CLAHE according to monitoring fields. Unfortunately, the parameters of CLAHE are usually fixed, which makes it difficult to adapt to various places in coal mines. Moreover, the GAN (Generative Adversarial Network) based image augmentation algorithms require huge computing resources leading to serious degradation of real-time performance [23]. Meanwhile, the datasets for training GAN are difficult to obtain in coal mines. Hence, GAN-based image augmentation algorithms are not suitable for coal mine real-time intelligence monitoring.

To solve the abovementioned problems, we proposed CAP-YOLO and AEPSM for coal mine real-time intelligent monitoring. First, DCAM (Deep Channel Attention Module) is designed for evaluating the importance level of channels. Then, we removed the filters corresponding to low-importance channels in YOLOv3 to form CAP-YOLO. Meanwhile, fine-tuning is used to recover the accuracy of CAP-YOLO. Finally, the AEPSM is designed and combined with the Backbone of CAP-YOLO, which has the ability to adaptively select parameters of CLAHE according to environments.

The main contributions of this paper are summarized as follows:(1)DCAM is designed for evaluating the importance level of channels in feature maps.(2)The coal mine pedestrian dataset was established for transfer learning YOLOv3. Then, the YOLOv3 was pruned with the guidance of DCAM for forming CAP-YOLO.(3)For the complex lighting environments in coal mines, AEPSM proposed and combined with the Backbone of CAP-YOLO to perceive the lighting environment, to set the parameters of CLAHE for improving the accuracy of object detection.

The remainder of this paper is organized as follows. Related methods about model pruning and attention mechanisms are introduced in Section 2. In Section 3, DCAM, AEPSM, and pruning approaches are proposed. Section 4 provides an experiment and comparison of the proposed approaches. Finally, we conclude this paper in Section 5.

## 2. Related Methods

### 2.1. Model Pruning

Model pruning is an important means to reduce the number of model parameters and calculations. An early study of model pruning was mainly undertaken to prune the weights, and the importance level of weights was evaluated by their magnitude [24,25]. This is a non-structured pruning, and it destroys the structure of CNN. Therefore, this approach reduces the number of parameters, but does not provide acceleration unless special hardware and libraries are used. Structured pruning, which focuses on finding and removing whole filters, is a hardware-friendly and coarse-grained method, and it has received a lot of attention recently. Hence, how to evaluate the importance level of channels or filters is a hot research topic of structured pruning. Hu, H. and Peng, R. [26] evaluated the importance level of filters using APoZ (Average Percentage of Zeros) of the output feature maps. They argue that the higher the APoZ the less important the filter is. Liu, Z. and Li, J. [27] thought that the scale factor of BN (Batch Normalization) reflects the importance of the corresponding filter. Therefore, they evaluated the importance level of filters using the L1-norm of scale factors. The proposed method completed a 20× reduction in model size and a 5× reduction in computing operations. Li, H. and Kadav [28] used the L1-norm of filter weights as the importance criteria for filters. The papers [29,30] proposed that the importance of features is related to its entropy, and the higher the value of entropy, the more information the filter outputs. Hence, they pruned the filters which have a low value of entropy. He, Y., Zhang. X. and Sun, J. [10] evaluated the importance level of filters using LASSO (least absolute shrinkage and selection operator) regression. Then, they reconstructed the network according to the least squares. The method they proposed accelerated the VGG-16 by five times, and only a 0.3% drop in accuracy was observed. Luo, J.H. et al. [6] thought that the importance level of filters is reflected in the output of the next layer. They used the greedy method to prune filters in training and inference stages, which reduced the size of the VGG-16 model to 5.05 MB, and the accuracy was only reduced by 1%. The aforementioned researchers have studied the importance identification methods for filters or channels. However, they did not further investigate the pruning process and methods.

For the problem that the importance level of filters/channels is difficult to evaluate, He, Y.H. and Han, S. [31] proposed an auto-pruning method using reinforcement learning. However, this method has complex model construction and requires a huge amount of calculation. Therefore, this approach is not applicable for object detection models with larger model sizes. The papers [7,32] evaluated the importance level of filters using the L2-norm of the weights of filters. The selected pruning filters were set to 0 and retraining, and then they evaluated the filters again. This method achieves an excellent pruning effect; however, the approach requires multiple training for models. Luo, J.H. and Wu, J. [33] have designed a model to achieve importance evaluation and pruning automatically based on deep learning. However, the end-to-end approach has difficultly converging in practical situations.

The abovementioned approaches prune CNN models from different perspectives, laying the foundation for researchers to prune more complex object detection models. The papers [5,8,34] evaluated the importance of filters in YOLOv3 using the scale factors of BN layers. Then, they remove the filters with low importance and use various optimization to recover the accuracy of models. Chen, Y., Li, R. and Li, R.F. [35] think that the larger shift factors of BN layers are more indicative of the importance of filters relative to the scale factors. They combine scale and shift factors to prune YOLOv3, which obtains a better pruning performance than that of SlimYOLOv3 [8]. Chen, S. et al. [36] use depth-width convolution to replace the traditional convolution in YOLOv3 at first. Then, they pruned YOLOv3 based on the value of scale factors of BN layers. Finally, the knowledge distillation is used to further compress the model and obtain the Tiny YOLO-Lite model. Although this model has the ability to detect SAR (Synthetic Aperture Radar) ships in real-time, it is difficult to deploy in coal mines because of the complex lighting environment. The papers [18,19,20,21] have deployed object detection models on embedded platforms using model pruning, which has been widely used in many fields such as industry, agriculture, and so on. The above approaches mainly use the scale factors of BN layers as the criteria of filter importance. However, the values of scale factors in the Backbone are generally larger than those of other parts when the pre-trained weights are transformed and learned for a special task, which limits the pruning ratio. Therefore, it is unreasonable to use the scale factors of BN layers to evaluate importance level of filters in model pruning.

### 2.2. Attention Mechanism

The attention mechanism can improve the importance information and suppress unnecessary information, to enhance the performance of models [11,12]. Therefore, the attention mechanism is widely used in NLP [37], image classification [13], and object detection [14] fields. Hu, J. et al. [15] proposed SENet (Squeeze-and-Excitation Networks) to model the channel-wise relationships of CNN and adjust the feature response values of each channel. However, SENet ignores the spatial attention in feature maps. The SGE (Spatial Group-wise Enhance) module [38] is proposed to enhance semantics information and suppress noise, but the channel attention is ignored. The BAM (Bottleneck Attention Module) and CBAM (Convolutional Block Attention Module) have been proposed in [39,40], respectively, and the experiments illustrate that combining spatial and channel attention is superior to using either of them. C. Tang et al. [41]. have proposed a Channel Attention Module to promise the effectiveness of DeFusionNET, but the Channel-downscaling is used in channel attention modules for dimensionality reduction, which causes information loss of the input features. X. Lu et al. [42] have improved the detection accuracy of SSD for small objects by combing spatial and channel attention. Xue et al. [43] have proposed MAF-YOLO (multi-modal attention fusion YOLO) based on a dual attention module, to obtain more information from small objects. The attention mechanism is also used to improve the performance of YOLOv4 [44]. To sum up, the attention mechanism improves the model performance, but it also results in the disadvantages of increasing computation and model size. This disadvantage makes the model optimized by attention mechanisms that are difficult to deploy on edge devices.

For the channels of feature maps in CNN, the more important the channels have, the larger the scale value of the attention mechanism outputs. Therefore, the importance level of channels in a model can be evaluated by attention mechanisms, which could guide the model pruning. Yamamoto et al. [45] proposed the channel attention module PCAS (Pruning Channels with Attention Statistics) for evaluating channel importance and pruning the filters corresponding to low-importance channels. SENet is also used as a criterion of channel importance for model pruning [46]. However, the construction of SENet and PCAS is mainly composed of MLP, which has a weaker image feature extraction ability than that of CNN. Meanwhile, feature dimensionality reduction in SENet and PCAS leads to the serious loss of information. Therefore, SENet and PCAS have difficulty extracting more channel information. The CASCA is proposed in [16]. The authors combine spatial and channel attention to identify the redundant channels. Compared with other approaches, this method achieves higher accuracy at the same pruning ratio. Shi, R. et al. [47] evaluate the redundant channels in YOLOv3-tiny by combing spatial and channel attention. They have pruned YOLOv3-tiny according to the scale value of the attention module and deployed the pruned model on embedded platforms. However, compared with YOLOv3, YOLOv3-tiny has low detection accuracy, especially for small objects. Hence, the performance YOLOv3-tiny limits the accuracy of the proposed model. Currently, the attention modules for evaluating channel importance are simple in design, and it is difficult to effectively analyze the importance level of each channel in the feature map. To address those problems, we propose a new attention module to evaluate channel importance and illustrate the advantages by comparing with other methods in the experiment.

## 3. Methods

### 3.1. Review of the YOLOv3 Object Detection Model

YOLOv3 [2] is a classical one-stage object detection model. The structure of YOLOv3 can be roughly divided into the Input, Backbone, Neck, and Predict parts, as shown in Figure 1. The Backbone is used to extract the features of the input image and is mainly composed of five Resblocks. The main function of the Neck is feature fusion and extraction, so that YOLOv3 obtains multi-scale detection ability. The Predict part is used to integrate the features of the Neck and output the location and classification of the objects. YOLOv3 has excellent real-time performance and accuracy compared with other object detection models [3,17], and the construction of YOLOv3 is simple but effective. Therefore, YOLOv3 is convenient for evolution and is already applied in various fields.

YOLOv3 has excellent accuracy and real-time performance, but the requirement of calculations makes it still difficult to deploy on edge devices. Hence, YOLOv3 only performs high real-time performance on advanced GPU platforms that have powerful computing ability. For the coal mine surveillance video processing, there is no need to detect multiple types of objects, so it will have sufficient generalization ability even if the filters are pruned. To this end, we designed a channel attention module to identify the redundant channels, and the filters corresponding to those channels were removed for improving the real-time performance and reducing the model size.

### 3.2. Deep Channel Attention Module (DCAM)

The channel attention module is used to perceive the importance level of each channel in the feature map, enhance the important channels, and suppress the redundant channels [16,41]. In previous works, some proposed attention modules such as BAM and CBAM integrate the spatial and channel attention for improving model performance. However, the fusion attention can interfere with channel importance evaluation. Moreover, completely prohibiting the information interaction between channels also led to the inability to perceive global information. In order to solve those problems and avoid dimensionality reduction such as SENet, the group convolutional and Group Normalization (GN) are used to extract features and normalization, respectively. Meanwhile, the replacement of BN by GN can eliminate the influence of batch size [16]. The channel attention module we proposed is named DCAM (Deep Channel Attention Module). The structure of DCAM is shown in Figure 2.

For the input feature map $X,(X\in R^{C\times W\times H})$, we perform the convolution operation with the kernel size of 3×3:(1)$$X_{c1}=f_{\mathrm{cov}1}(X)$$
where $X_{c1},(X\in R^{C\times W\times H})$, *C* is the number of channels, *W* and *H* are the size of feature maps, $f_{\mathrm{cov}}$ denotes convolution operation.

Then, the maximum pooling and average pooling are used to extract features of $X_{c1}$ from two perspectives:(2)$$X_{\max1}=f_{\max pool}(X_{c1})$$
(3)$$X_{avg1}=f_{avgpool}(X_{c1})$$
where $X_{\max1}\in R^{C\times H/2\times W/2}$, $X_{avg1}\in R^{C\times H/2\times W/2}$. The maximum pooling is used to extract the important information while the average pooling is adopted to extract the global information.

We use the convolution operation with the kernel size of 1×1 to integrate $X_{\max1}$ and $X_{avg1}$, and the fusion feature is normalized by the GN layer:(4)$$X_{c2}=GN(f_{\mathrm{cov}2}(X_{\max1})+f_{conv3}(X_{avg1}))$$
where $X_{c2}\in R^{C\times H/2\times W/2}$. We set the GN layer with groups = 4, which causes the module to have a certain ability of information interaction in channels.

We further extract the features of $X_{c2}$ by convolution operation:(5)$$X_{c3}=f_{\mathrm{cov}4}(X_{c2})$$
where $X_{c3}\in R^{C\times H/2\times W/2}$. $X_{c3}$ is half the size of the input feature map, but the number of channels is still *C*. After multiple feature extraction, the information of the input feature map is compressed into $X_{c3}$.

The adaptive maximum pooling and adaptive average pooling are used to integrate the features of $X_{c3}$, while the GN layer is used to normalize the features:(6)$$X_{c4}=GN(f_{\max pool}(X_{c3}))+GN(f_{avgpool}(X_{c3}))$$
where $X_{c4}\in R^{C\times 1\times 1}$. The size of $X_{c4}$ is C×1×1, and the most representative features in each channel of the input feature maps are extracted in $X_{c4}$.

The information of $X_{c4}$ is extracted by convolution operation, and the features are normalized by GN. Finally, the scale value of DCAM is output by the active function sigmoid:(7)$$A_{C}=\sigma (GN(f_{cov5}X_{c4}))$$
where σ represents the sigmoid function, $A_{C}\in R^{C\times 1\times 1}$. Each element in $A_{C}$ represents the importance level of its corresponding channel.

DCAM uses multiple convolution and pooling to obtain deeper features of the input channels. The computational process of DCAM is more complex than other channel attention modules. However, the DCAM is not computationally intensive because of the use of group convolution. Meanwhile, the DCAM is only used to evaluate the importance level of channels, and it will be removed after pruning the model. Therefore, DCAM does not affect the real-time performance.

### 3.3. CAP-YOLO (Channel Attention Based Pruning YOLO)

BN has the effect of improving generalization ability and accelerating the convergence of the training process. Hence, BN is widely used in object detection models. The scale factors of BN represent the importance of the corresponding filters to a certain extent. Meanwhile, the value of scale factors was conveniently obtained. Therefore, the pruning methods based on BN scale factors are easy to implement. However, when coal mine pedestrian dataset is used for transfer learning YOLOv3 based on the pre-trained weights, the BN scale factors of Neck is significantly less than that of Backbone. The phenomenon is shown in Figure 3: the values of uneven distribution in BN scale factors seriously constrain the maximum pruning ratio (the computing method of the maximum pruning ratio is shown below).

Evaluating the importance level of channels by DCAM avoids the influence of pre-trained weights on maximum pruning ratio, because the training process of DCAM and the pre-trained weights are separated. When DCAM is trained, the weights of YOLOv3 are fixed, and only the parameters of DCAM are gradually trained from the initial state to the convergent state for minimizing loss function. Therefore, the computational power required to train the DCAM is lower than training the entire model.

For preventing over-pruning, the maximum pruning ratio is computed by:(8)Plimit=arg(Im(min(Im(l)max))sort)/N
where Im(l)max represents the maximum importance value of the filters or channels in the *l*-th convolution layer; Im()sort denotes the list of Im which is listed from small to large; *N* is the total number of evaluated channles. The larger the Im(l)max, the more important the filter or channel is. The main function of $P_{\mathrm{limit}}$ is to prevent removing all filters of a convolution layer.

For YOLOv3, we primarily prune the Backbone and Neck. Resblock is the main component of Backbone. Before pruning the Resblock, we insert DCAM behind the first convolutional layer of the Resblock to form the Res-attention module as shown in Figure 4. The DCAM evaluates the channel importance level of the first convolutional layer according to the output feature maps.

The DCAM is inserted into YOLOv3 to form the YOLO-DCAM. Then, we trained YOLO-DCAM after fixing the pre-trained weights of YOLOv3. After training convergence, the images of the training set are inferred, and the average of the output of DCAM is calculated as the importance level of channels:(9)Im(l,j)=1D∑i=0DDCAMl,j(i)
where *j* denotes the *j*-th channel in *l*-th later. *i* represents the *i*-th image of the training set, *D* denotes the training set.

The pruning percentage $p_{tr}$ ($p_{tr}<p_{\mathrm{limit}}$) is set according to the evaluation result, and the pruning threshold is calculated by:(10)pth=Im(ptr×N)sort
when the value of channel importance corresponding to a filter is less than $p_{th}$, the filter is removed. Moreover, with the pruning of a layer, the output channels in the feature map of this layer are reduced. Therefore, the corresponding channels of the filters in the next layer also need to be removed. The pruning process of the *l*-th layer is shown in Figure 5.

The pruning process is summarized as follows. Firstly, the DCAM is inserted into YOLOv3 to form YOLO-DCAM. Secondly, the pre-trained weights of YOLOv3 in YOLO-DCAM are fixed, then we train the YOLO-DCAM until convergence. Thirdly, the importance level of channels is evaluated by (8), and the filters corresponding to the low importance are removed to form CAP-YOLO. Finally, we fine-tune the CAP-YOLO for recovering accuracy. The pruning process is shown in Algorithm 1.
**Algorithm 1:** Pruning Process1Initialize YOLO-DCAM2Load the parameters of YOLOv3 to YOLO-DCAM3Fix the YOLOv3’s parameters of YOLO-DCAM4Training YOLO-DCAM5for img to D: Im(l,j)=Im(l,j)+SCAMl,j(img)
6for l to L: //L is the number of total layers of pruning layers in YOLOv3Im(l)max=max(Im(l,j))
7Get the maximum prune value plimit=min(Im(l)max)
8Set prune threshold $p_{tr}$$(p_{tr}<p_{\mathrm{limit}}$)9Get CAP-YOLO by prune $filters_{i,j}$
 whose Im(l,j)<ptr
10Fine-tune the CAP-YOLO

### 3.4. Adaptive Image Enhancement Parameter Selection Module

The lighting environment in different monitoring areas of a coal mine is different. There is sufficient lighting in some places, while the lighting in some areas is insufficient or uneven. This phenomenon makes the accuracy of models vary from area to area. CLAHE is a classical image augmentation algorithm widely used in the field of image analysis. However, the fixed parameters of CLAHE are difficult to adapt to all areas in a coal mine. The effect of CLAHE with different parameters is shown in Figure 6.

From Figure 6, it can be seen that for the CLAHE, the different parameters lead to different effects for a same image. Therefore, we proposed AEPSM (Adaptive image Enhancement Parameter Selection Module), which adaptively adjusts the parameters of CLAHE by perceiving the lighting environment, to improve the accuracy of CAP-YOLO in various fields of a coal mine. AEPSM further processes the image features extracted by CAP-YOLO’s Backbone to output the best parameters of CLAHE under the current lighting environment. The structure of AEPSM is shown in Figure 7.

AEPSM is inserted into CAP-YOLO. In the process of training, the weights of CAP-YOLO are fixed, and only the parameters of AEPSM are trained. The loss function of the training processing is the same as YOLOv3 [2]. In the process of inference, the parameters of CLAHE are generated by APESM and Backbone at first. Then, the parameters are fixed to this environment and the AEPSM is ignored in the inference process to save computing resources. The training and inference processes of AEPSM are shown in Figure 8.

## 4. Results

### 4.1. Experiment Environments

#### 4.1.1. Software and Hardware Environments

The hardware and software environments used in the experiment are shown in Table 1.

#### 4.1.2. Dataset

The dataset of COCO is used to train and evaluate our method at first. Meanwhile, the coal mine pedestrian dataset is built for further training and evaluation of CAP-YOLO and AEPSM. The coal mine pedestrian dataset includes 10 monitoring areas of the coal mine, and each area has 600 images, for a total of 6000 images. We divide 4000 pictures in the dataset into training sets and 2000 pictures into evaluation sets. For improving the generalization ability of the model, the dataset is extended by flipping, cropping, and adding Gaussian noise, as shown in Figure 9.

#### 4.1.3. Details

We follow the default configuration of Darknet to train YOLO-DCAM and CAP-YOLO. The size of the input images is set to 416×416. We set SGD as the optimizer with the momentum = 0.9 and weight_delay = 0.05. The initial learning rate is 0.001, and the decay factor is set to 0.5, which decays the learning rate per 1000 steps in 30,000–50,000 iterative steps.

### 4.2. Performance on COCO

The proposed DCAM is used to distinguish the important channels and redundant channels in YOLOv3. Meanwhile, the performance of YOLOv3 can be improved by DCAM, due to the DCAM’s ability of enhancing the important channels. In order to evaluate the DCAM’s ability of feature enhancement and identify important channels, the SENet, SGE, BAM, and CBAM (the CBAM is set to extract spatial attention first, and then extract the channel attention, the two attentions combined in a sequential manner) are inserted into YOLOv3 to form YOLO-SENet, YOLO-SGE, YOLO-CBAM, YOLO-BAM, which are compared with YOLO-DCAM and YOLOv3, SSD on the COCO dataset. The results of the comparison are shown in Table 2.

As shown in Table 2, the accuracy of YOLO-DCAM is lower than YOLO-CBAM and YOLO-BAM, but higher than YOLO-SGE and YOLO-SENet. The results show that DCAM has better important channel identifying and enhancing abilities than SENet. Compared with CBAM and BAM, the important information enhancing ability of DCAM is weak, due to the less frequent communication between channels in DCAM. Therefore, the ability of DCAM in spatial information enhancement is lower than that of CBAM and BAM. However, the main function of DCAM is to evaluate the importance level of channels and guide the pruning process. Then, the different channel evaluation approaches are used to guide the pruning process on YOLOv3, and the effects of those pruning methods are validated based on COCO.

We compare the CAP-YOLO with the following pruning methods: (1) Slim-YOLO [15]. (2) The pruning model is based on SENet, which named YOLO-SENet-prune. (3) The pruning model is based on BAM, which is named YOLO-BAM-prune. (4) The pruning model is based on CBAM, which is named YOLO-CBAM-prune. The mAP of each model under various pruning ratios is shown in Figure 10.

It can be seen from Figure 10, with the increasing pruning ratio, the accuracy of all models is reduced. The reason for this phenomenon is that model pruning leads to a reduction in the ability of feature extraction and generalization. However, because the results of evaluating the importance level of channels vary from evaluation approach to approach, the accuracy of the model obtained using different pruning approaches is also different even if the pruning ratio is the same. From Figure 10, the accuracy of the pruning model based on attention mechanisms is higher than that based on BN scale factors (Slim-YOLO) with the same pruning ratio. This is because the attention module, which contains neural networks, fits the importance level of each channel in training. The convergence process is guided by the loss function. Meanwhile, although the BN scale factors reflect the importance level of filters to some extent, it is difficult to represent the importance level of channels.

The effect of model pruning based on DCMA, BAM, and CBAM is better than that based on SENet, because the SENet greatly reduces the dimensions of features, which results in serious information loss. DCAM-based model pruning is superior to that of BAM and CBAM, the reason is that the DCAM does not fuse the channel features for preventing the interference between channels. Meanwhile, a little communication between channels by GN enables DCAM to perceive global information.

In order to establish the real-time performance of CAP-YOLO, we set the pruning ratio to 50%, 70%, and 88% (the maximum pruning ratio of CAP-YOLO is 88.7%) to compare with YOLOv3-tiny and YOLOv3; the result is shown in Table 3.

It can be seen in Table 3, when the pruning ratio is set to 88%, that the mAP of CAP-YOLO is still maintained at 39.8% on COCO. The accuracy of CAP-YOLO under the maximum pruning ratio is higher than that of YOLOv3-tiny, the inference speed is faster than that of YOLOv3, and the weight size of CAP-YOLO is only 28.3 MB, which is also less than that of YOLOv3-tiny.

### 4.3. Performance on the Coal Mine Pedestrian Dataset

#### 4.3.1. Performance of CAP-YOLO on the Coal Mine Pedestrian Dataset

Compared with COCO, there are only pedestrians in the coal mine pedestrian dataset. For neural network models, the fewer the classifications processed, the lower the generalization capability required. On the coal mine pedestrian dataset, we compared the DCAM, BAM, CBAM, SENet, and BN scale-factor-based pruning methods. The pruning process is the same as the previous part, and the results are shown in Figure 11.

It is shown in Figure 11 that the DCAM-based pruning achieves better accuracy than that of other approaches. Meanwhile, the maximum pruning ratio of attention-based pruning methods is higher than the BN scale-factor-based approach (the reason is explained in Section 3.3). The performance of accuracy and real-time under the maximum pruning ratio is shown in Table 4.

It can be seen in Table 4 that the maximum pruning ratio of CAP-YOLO is 93%, while maintaining 86.7% mAP. The speed and accuracy are all superior to that of other pruning methods. YOLOv3-tiny is a simplified version of YOLOv3, the speed of YOLOv3-tiny is faster than other methods, but the accuracy is the lowest because of the weak feature extraction ability. The inference speed of CAP-YOLO is improved by increasing the pruning ratio for removing redundant filters, and the speed of CAP-YOLO reached 31 FPS on the embedded platform of NVIDIA Jetson TX2. To sum up, although there is a small loss in accuracy, the CAP-YOLO, which was pruned based on DCAM, greatly improved the real-time performance and reduced the model size. Therefore, the superior real-time performance and accuracy of CAP-YOLO enable it to be deployed on edge devices for coal mine real-time intelligence monitoring.

#### 4.3.2. Performance of AEPSM

AEPSM is used to adaptively select the parameters of CLAHE for image augmentation under different fields. For the coal mine pedestrian dataset, we first test the CAP-YOLO (93%) using 10 field images. Then, the parameters of CLAHE are fixed to clipLimit = 2.0, tileGridSize = (8, 8), which enhances the images before input to CAP-YOLO. The results of test accuracy on 10 fields are shown in Table 5.

From Table 5, with the application of CLAHE, the detection accuracy of CAP-YOLO in some fields has increased, while the accuracy in some fields has decreased. The reason for this result is that the fixed parameters of CLAHE are not adapted in every field.

The training process of AEPSM is shown in Figure 8a. After training convergence, AEPSM is inserted into CAP-YOLO as shown in Figure 8b to select parameters for different fields. Finally, the parameters of CLAHE selected by AEPSM are used to augment the input images for CAP-YOLO, as shown in Figure 8c. AEPSM only needs to infer once for parameter selection when deployed on a new field, and then it will not be inferred to save computation resources. The test result of CAP-YOLO with the AEPSM-CLAHE is shown in Table 6.

From Table 6, the AEPSM set different parameters for CLAHE in different fields. Compared with the fixed parameters, AEPSM adaptively sets parameters by perceiving the lighting environment in different fields so that CAP-YOLO can obtain better detection accuracy. The effect of CLAHE with AEPSM is shown in Figure 12.

## 5. Discussion

Channel attention has the ability to identify and enhance the important channels in feature maps. Therefore, the more important the channels of feature maps are, the higher the scale value of the DCAM outputs. According to this phenomenon, DCAM has the ability to evaluate the importance level of channels and identify the redundant filters. Hence, the CAP-YOLO retains an mAP of 86.7% while the pruning ratio has reached 93%. Compared to the traditional parameter setting methods, AEPSM set the parameters of CLAHE adaptively based on the lighting environments of fields, so that the CAP-YOLO can obtain higher accuracy for different lighting environments.

## 6. Conclusions

In this paper, the DCAM was proposed to evaluate the channel importance level and identify the redundant channels; then, we pruned YOLOv3 based on DCAM to form CAP-YOLO. CAP-YOLO reached 86.7% mAP when the pruning ratio was set to 93% and achieved 31 FPS inference speed on NVIDIA Jetson TX2. Meanwhile, we further proposed AEPSM to perceive the lighting environments of different coal mine fields, which adaptively set the parameters of CLAHE for improving the accuracy of CAP-YOLO.

In the future, we will undertake a further study on channel attention mechanisms for evaluating the importance level of channels. In addition, we will design a special loss function or optimization method for DCAM and CAP-YOLO in the next step, for improving the real-time performance and accuracy of intelligent video monitoring.

## Figures and Tables

**Figure 1 sensors-22-04331-f001:**
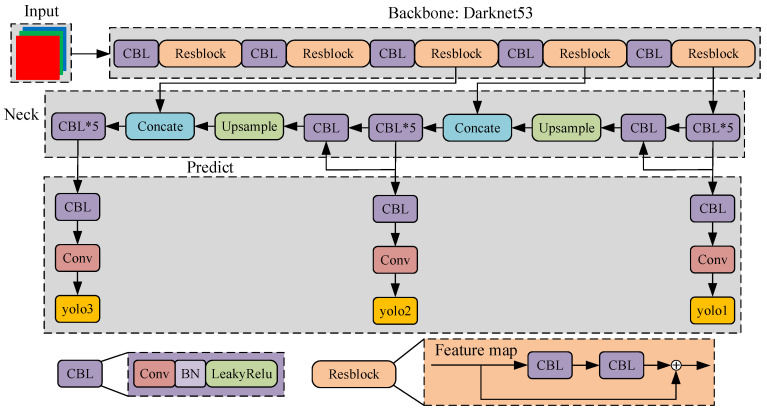
Structure of YOLOv3, where the CBL represents the combination of Convolutional, Batch Normalize, and LeakyRelu activation function. Resblock denotes the residual structure. The outputs of YOLOv3 are represented by yolo1, yolo2, and yolo3, which represent different output scales.

**Figure 2 sensors-22-04331-f002:**
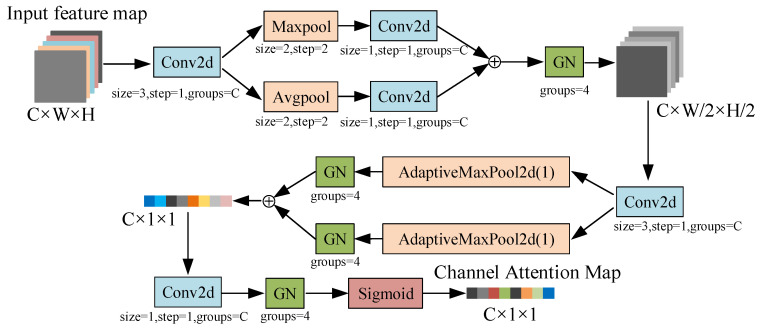
Structure of DCAM.

**Figure 3 sensors-22-04331-f003:**
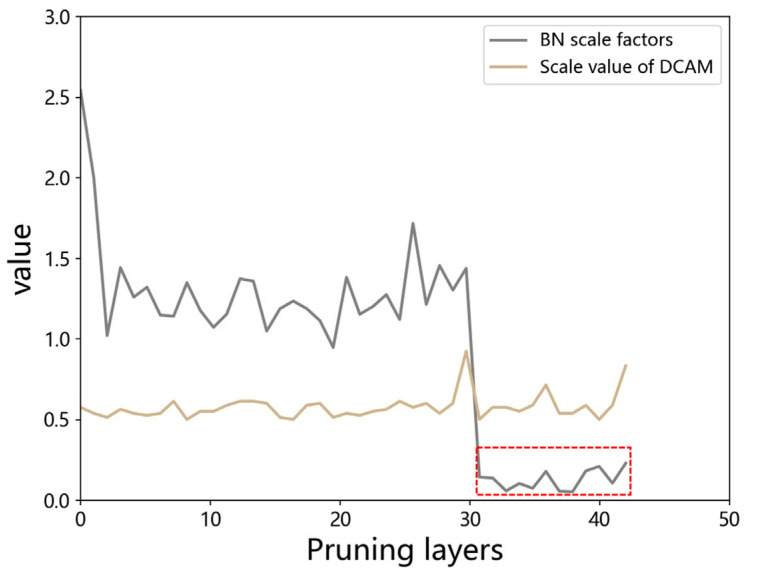
Distribution of importance values evaluated by BN scale factors and DCAM. Most of the convolution layers in the Neck part are contained in the red dotted box, and the values of those are significantly smaller than that of Backbone. Obviously, the importance values evaluated by DCAM are more evenly distributed.

**Figure 4 sensors-22-04331-f004:**
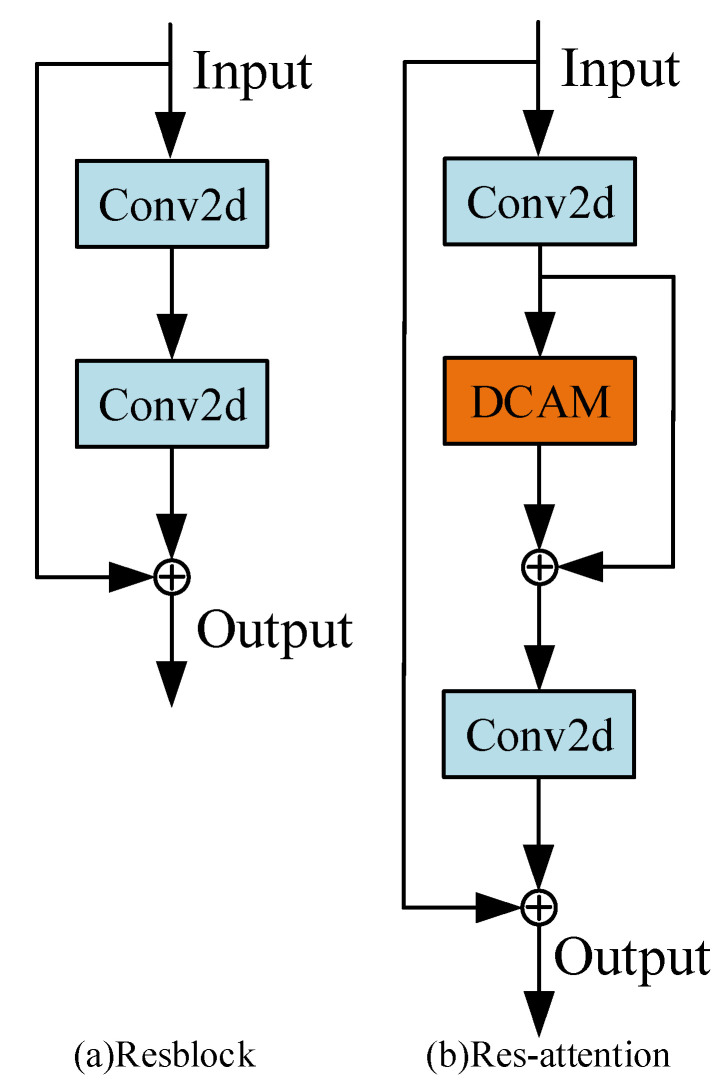
(**a**) Structure of Resblock in Backbone. (**b**) Structure of Res-attention.

**Figure 5 sensors-22-04331-f005:**
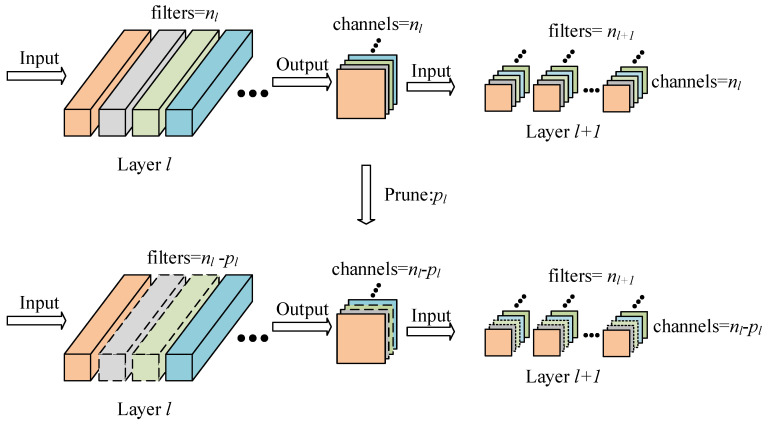
Pruning process of the *l*-th layer, where the $n_{l}$ denotes the number of filters in the *l*-th layer, while $p_{l}$ represents the number of the filters that were removed.

**Figure 6 sensors-22-04331-f006:**
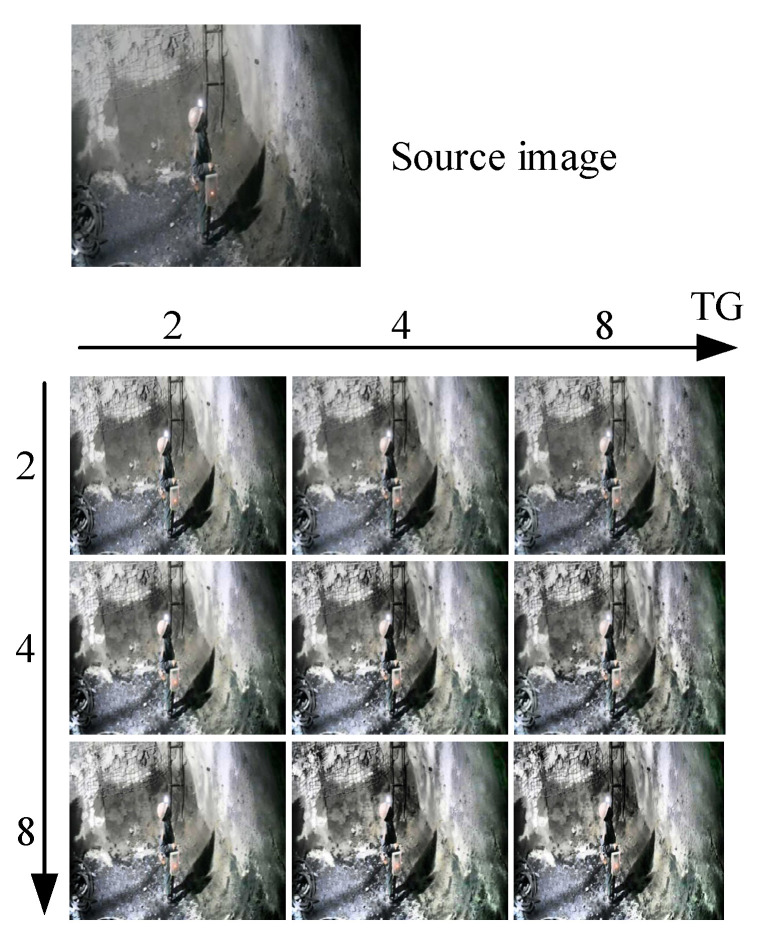
Effect of CLAHE with different parameters, where the parameter TG represents the tileGridSize, which indicates how many parts the image will be segmented into. The parameter of CL denotes clipLimit, which is the limit value of clips in CLAHE.

**Figure 7 sensors-22-04331-f007:**
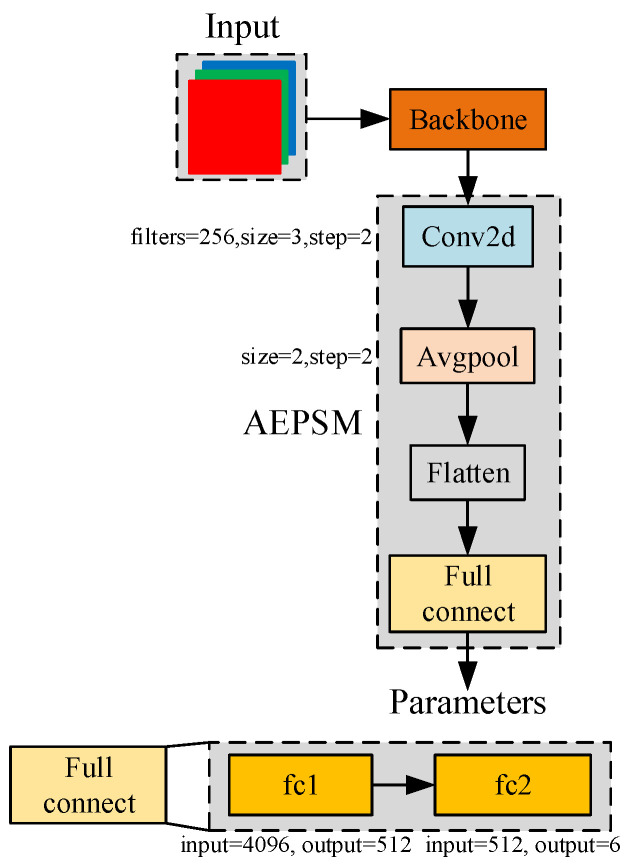
Structure of AEPSM. The output dimension of AEPSM is 6: the former 3 correspond to clipLimit = 2, clipLimit = 4, and clipLimit = 8, respectively, while the later 3 correspond to tileGridSize = 2, tileGridSize = 4, and tileGridSize = 8.

**Figure 8 sensors-22-04331-f008:**
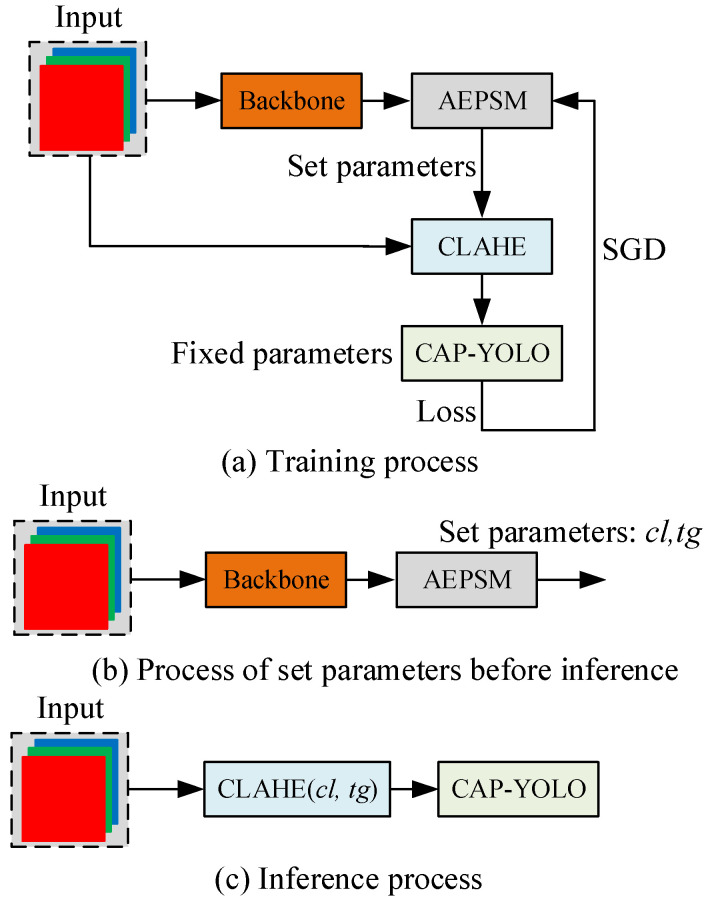
Training and inference processes of AEPSM. (**a**) is the training process, (**b**) is the parameter selection process, and (**c**) is the inference process.

**Figure 9 sensors-22-04331-f009:**
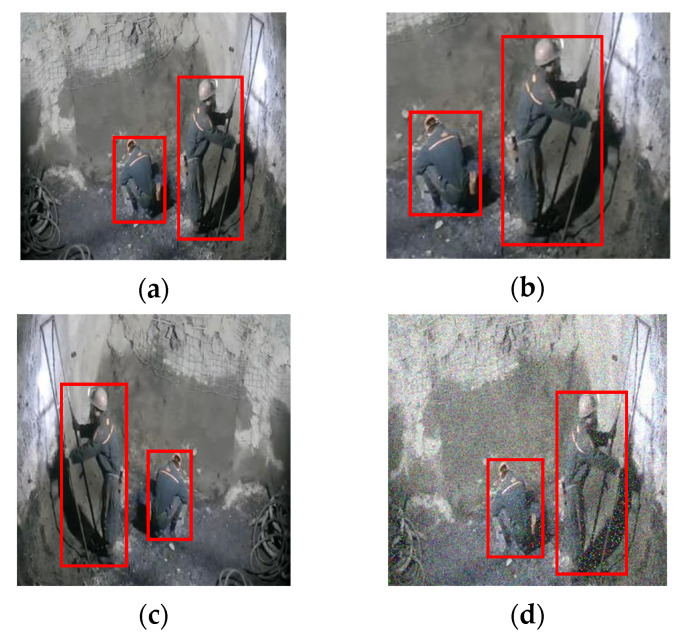
Effect of image modification. (**a**) Source, (**b**) cropping, (**c**) flipping, and (**d**) adding Gaussian noise.

**Figure 10 sensors-22-04331-f010:**
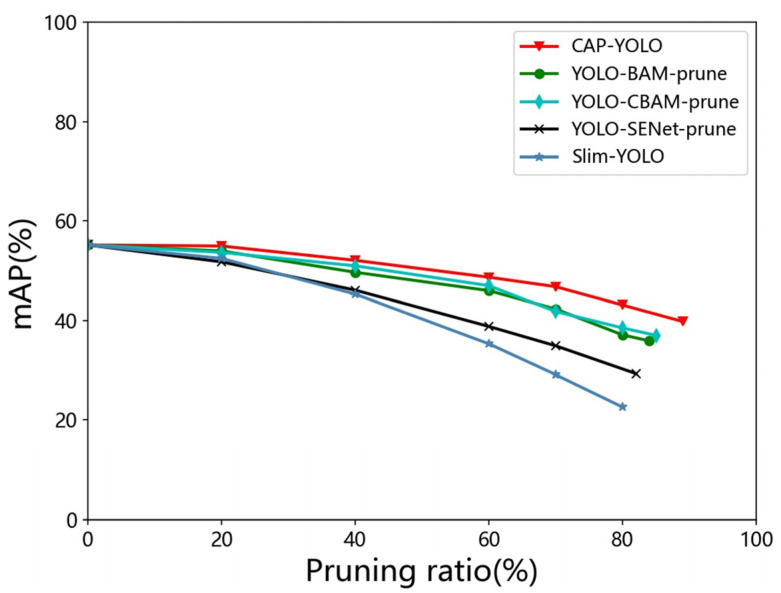
Pruning ratio-mAP curve of each model.

**Figure 11 sensors-22-04331-f011:**
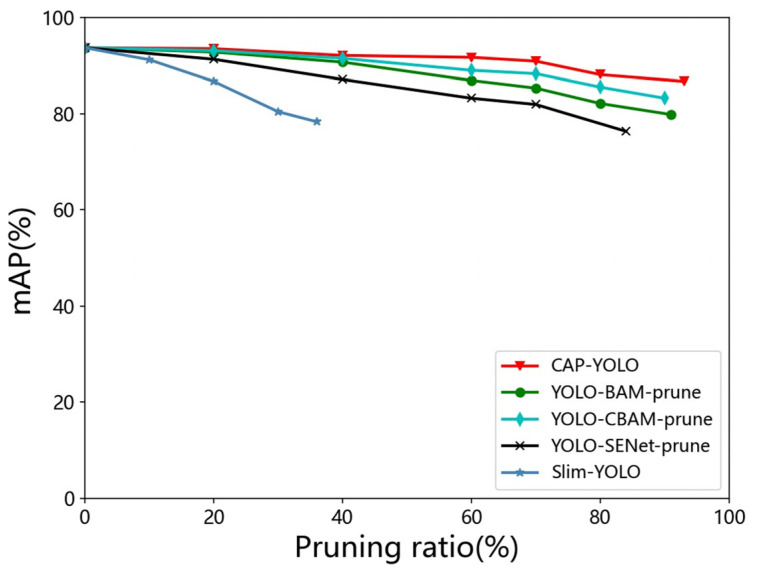
Pruning ratio-mAP curve on the coal mine pedestrian dataset.

**Figure 12 sensors-22-04331-f012:**
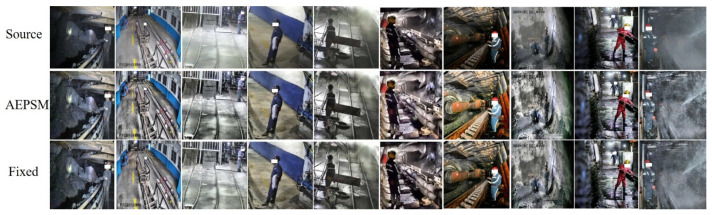
The effect of CLAHE with AEPSM and fixed parameters under different fields.

**Table 1 sensors-22-04331-t001:** Experiment environments.

Training and Deployment Platform	Embedded Platform
Intel i7-11700k @4.9GHz	NVIDIA Jetson TX2
NVIDIA RTX3090	Ubuntu 18.04
RAM 64G	Python3.6.8
Ubuntu 18.04	Pytorch 1.10
Python3.6.8	CUDA 11.3
Pytorch 1.10	
CUDA 11.3	

**Table 2 sensors-22-04331-t002:** Comparison results of YOLO-DCAM, YOLO-SENet, YOLO-SGE, YOLO-CBAM, YOLO-BAM, YOLOv3, and SSD.

Model	mAP (%)	FPS (RTX3090)
YOLO-DCAM	62.3	48
YOLO-SENet	58.7	52
YOLO-SGE	59.4	51
YOLO-CBAM	66.8	55
YOLO-BAM	64.1	57
YOLOv3	55.2	61
SSD	50.1	63

**Table 3 sensors-22-04331-t003:** Comparison of various models.

Model	mAP (%)	FPS-GPU	Size (MB)	FLOPs (Bn)
CAP-YOLO (40%)	52.1	87	127	35.04
CAP-YOLO (60%)	48.7	109	86.4	25.32
CAP-YOLO (88%)	39.8	182	28.3	7.38
YOLOv3-tiny	33.1	173	33.1	5.56
YOLOv3	55.2	61	236	65.86

**Table 4 sensors-22-04331-t004:** Performance of accuracy and real-time under maximum pruning ratio.

Model	mAP (%)	FPS-GPU	FPS-TX2
CAP-YOLO (40%)	92.1	87	12
CAP-YOLO (60%)	91.7	109	16
CAP-YOLO (93%)	86.7	171	31
YOLO-SENet-prune (84%)	76.3	154	23
YOLO-CBAM-prune (90%)	83.2	161	28
YOLO-BAM-prune (91%)	79.8	166	29
Slim-YOLOv3 (36%)	78.3	78	9
YOLOv3	93.7	61	6
YOLOv3-tiny	56.4	173	34

**Table 5 sensors-22-04331-t005:** Results of test accuracy on 10 fields.

Fields	mAP (CAP-YOLO)	mAP (CAP-YOLO + CLAHE)
1	90.1	88.0
2	85.9	89.6
3	88.4	83.9
4	89.6	88.1
5	89.3	87.3
6	82.8	80.8
7	81.5	84.2
8	85.9	88.7
9	87.1	90.2
10	86.4	88.5

**Table 6 sensors-22-04331-t006:** Test results of CAP-YOLO with AEPSM-CLAHE.

Fields	mAP (AEPSM + CLAHE)	CL	TG
1	91.6	4	8
2	91.4	2	8
3	92.1	8	4
4	90.7	4	4
5	91.6	2	8
6	90.1	2	4
7	92.3	8	8
8	92.9	8	4
9	91.8	2	8
10	92.6	4	8

## Data Availability

Not applicable.

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
