# Peer review of "CAP-YOLO: Channel Attention Based Pruning YOLO for Coal Mine Real-Time Intelligent Monitoring"

_sensors, 2022, doi:10.3390/s22124331_

Round 1
Reviewer 1 Report
The paper introduces channel attention-based pruning for YOLOv3, which reduces the number of channels by evaluating the importance of each channel with attention. In addition, the paper proposes the adaptive image enhancement parameter selection module to reflect different lighting environments.
The strengths of the paper are:
- The motivation is clear.
The paper clearly defines the target problem and addresses the challenges in coal mine monitoring (e.g., divergent lighting environments). I think it would also be good to add sample images in different lighting conditions to the introduction. - The technical approach seems proper and sound.
The weaknesses of the paper are:
- The writing quality of the paper is low.
The paper has too many grammatical errors and typos, which significantly degrade its readability. For example, "Then, the monitoring videos are analyzed by AI models in cloud server and return the results of video analysis through networks" in the introduction is grammatically wrong. I cannot name every error, but the authors should fix them in the revision.
The paper often uses technical terms without explaining them first. For example, the paper refers to CLAHE in the introduction and APoZ in the related method section without explaining them. It would be nice to briefly introduce the technical terms and give a full name before using an abbreviation (e.g., Contrast Limited AHE (CLAHE)). - The explanation and analysis of the experimental results are insufficient.
First, in Section 4.2, it is unclear how YOLO-SENet, YOLO-SGE, YOLO-CBAM, and YOLO-BAM were trained. Have the same training parameters been used for the models? Is it possible to show that they are the optimal parameters for the models? The parameters could be favorable to YOLO-DCAM.
Second, in Section 4.3.2, if AEPSM improved the accuracy only for half of the fields, I think it is not easy to conclude that AEPSM is effective. Therefore, a more strong conclusion would be crucial for AEPSM.
Other comments:
- The paper often uses "SCAM" when "DCAM" has to be used, such as in Figure 5.
- What does "p_tr x Σ (i, j)" mean in Equation 10? I cannot interpret the equation.
- The claim in Lines 188-189, "However, the construction of SENet and PCAS is too simple to extract more channel information." needs justification. Please add more explanation to support the claim. (Same for the claim in Lines 194-195)
- Please be consistent in citing a paper. The paper uses "First Name, Last name", "First Initial. Last name", and others alternately (e.g., Lines 115 and 117). In addition, please write only the first author's name when using "et al." (e.g., Xue at el. [42]).
Reviewer 2 Report
The paper proposes a method to prune channels of YOLOv3. The results are for COCO and a 5-class dataset.
It is difficult to verify the accuracies reported in the results section. Is the code available?
How is your algorithm for model punning guided by channel attention different from [16]. Applied to image classification or object detection, the approach seems to be similar.
In line 214, the authors state that “size makes it still difficult to deploy …”. Why? Considering your problem and target device, YOLOv3 can run on the device and maybe achieve the required timing constraints.
In section 3.3, the authors mention that “the phenomenon seriously constraint …”. This should be better explained.
The paragraph in line 282 is confusing. Please rewrite it.
In line 287, explain what is over-pruning.
In line 298, you explain the utilization of DCAM. After training, DCAM is removed, right? Do you remove it before fine-tuning? Please clarify.
The first paragraph of section 4 should be removed.
The pruning steps from lines 398-406 were already explained. At least, it should have already been explained. I suggest removing this from here.
Table 3 should include the number of operations to infer the complexity of the algorithm.
How do you explain the differences in the ratios of FPS in table 4 between GPU and eGPU?
Reviewer 3 Report
In order to deploy models on edge devices for real-time coal mine intelligent monitoring, this paper proposes CAP-YOLO (Channel Attention based Pruning YOLO) and AEPSM (Adaptive image enhancement parameter selection module). Experiment results demonstrate the efficacy and lightweight of the proposed network. Following issues should be addressed to improve this version:
1. In Figure 1. Structure of YOLOv3 , the abbreviations used should be explained in detail in the main body of text.
2. Channel attention module is widely used to perceive the important level of each channels of feature maps. The authors should discuss the difference between the proposed DCAM and previous channel attention module, such as: DeFusionNET: Defocus Blur Detection via Recurrently Fusing and Refining Discriminative Multi-scale Deep Features, PAMI 2022.
3. In addition to using channel attention, why not use spatial attention?
Round 2
Reviewer 2 Report
My issues have been clarified. The work can be published.